# Neuroendocrine Response and State Anxiety Due to Psychosocial Stress Decrease after a Training with Subject’s Own (but Not Another) Virtual Body: An RCT Study

**DOI:** 10.3390/ijerph19106340

**Published:** 2022-05-23

**Authors:** Dalila Burin, Gabriele Cavanna, Daniela Rabellino, Yuka Kotozaki, Ryuta Kawashima

**Affiliations:** 1Institute of Development, Aging and Cancer (IDAC), Tohoku University, 4-1 Seiryocho, Aobaku, Sendai 980-8575, Japan; ryuta.kawashima.a6@tohoku.ac.jp; 2Smart Aging Research Center (SARC), Tohoku University, 4-1 Seiryocho, Aobaku, Sendai 980-8575, Japan; cavanna.gabriele.t8@dc.tohoku.ac.jp; 3Graduate School of Science, Tohoku University, 6-3 Aramaki Aza-Aoba, Aobaku, Sendai 980-8578, Japan; 4Department of Psychiatry, Western University, 550 Wellington Rd., London, ON N6C 5J1, Canada; drabelli@uwo.ca; 5Division of Clinical Research and Epidemiology, Iwate Tohoku Medical Megabank Organization, Iwate Medical University, 1-1-1, Idaidori, Yahaba, Iwate, Morioka 028-3694, Japan; kotoyuka@iwate-med.ac.jp

**Keywords:** psychosocial stress, immersive virtual reality training, body ownership, agency, salivary alpha-amylase, anxiety

## Abstract

Previous research involving healthy participants has reported that seeing a moving virtual body from the first person perspective induces the illusion of ownership and agency over that virtual body. When a person is sitting and the virtual body runs, it is possible to measure physiological, behavioral and cognitive reactions that are comparable to those that occur during actual movement. Capitalizing on this evidence, we hypothesized that virtual training could also induce neuroendocrine effects that prompt a decreased psychosocial stress response, as occurs after physical training. While sitting, 26 healthy young adults watched a virtual avatar running for 30 min from the first person perspective (experimental group), while another 26 participants watched the virtual body from the third person perspective (control group). We found a decreased salivary alpha-amylase concentration (a biomarker for the stress response) after the virtual training among the experimental group only, as well as a decreased subjective feeling of state anxiety (but no difference in heart rate). We argue that the virtual illusion of a moving body from the first person perspective can initiate a cascade of events, from the perception of the visual illusion to physiological activation that triggers other biological effects, such as the neuroendocrine stress response.

## 1. Introduction

While a moderate amount of stress may be beneficial for individuals to adapt and develop, psychosocial stress (i.e., social evaluation, social exclusion and goal-directed performance judgment that is perceived as a social threat) is considered detrimental for overall well-being and can accelerate ageing-related disorders on several levels [1]. Whereas physical activity (aerobic or weight training, for example) is commonly considered to be an effective direct strategy for preventing illness and promoting bodily and mental health by reducing risk factors, such as cardiovascular diseases [2], it has also been suggested that physical activity can act indirectly via stress-buffering effects [3]. According to the cross-stressor adaptation theory, which is currently the most well acknowledged hypothesis that explains the connection between physical activity and stress reduction, being exposed to physical stress, such as moderate or vigorous exercise, elicits a stress response that is similar to the motoric fight-or-flight reaction, which activates the hypothalamic–pituitary–adrenocortical (HPA) axis and the autonomous nervous system (ANS). The HPA axis and ANS then regulate the physiological response in order to return to the baseline activation, thereby prompting the rewarding process and the subjective and behavioral feeling of coping with the stress [4]. By contrast, during psychosocial stress, the focus is on emotional regulation and goal-directed behavior, which discourage the rewarding process [1]. In this regulation system, the ANS and HPA axis control cardiovascular changes and the neuroendocrine response; in fact, heart rate variability, hormones and enzymes (such as cortisol, alpha-amylase or catecholamine) are typically used as measurements for the psychosocial stress response, in addition to subjective types of evaluations [5].

Despite the above-mentioned benefits of stress and even though physical activity is easily accessible, cost effective and rarely shows undesirable side effects, many people cannot perform strength or aerobic training (e.g., elderly people in frail conditions, people with neurological or physical disorders, severely obese people, patients recovering from hospitalization or chronic diseases, etc.). Unsurprisingly, people with lower fitness levels (such as those in the above-mentioned categories or very busy workers and sedentary people) are at a higher risk of cardiovascular diseases, stroke, metabolic syndromes and cognitive decline [6,7]. Therefore, young healthy subjects who do not show signs of frailty and who are perfectly capable of performing exercise may also be directly involved.

One ideal solution consists of bypassing the actual execution of movement while maintaining the same benefits for bodily and cognitive functions that are typically connected to physical activity [8]. Here, we propose to achieve this goal using immersive virtual reality (IVR), which is an innovative technique that allows the representation of a plausible and realistic virtual world and a virtual body (or avatar) using an IVR visor. Through IVR and, in particular, the use of visual perspective [9], it is possible to induce the illusion of owning and moving a virtual body when there is a mismatch between executed and seen movements [10] or even when the person is not moving at all [11,12,13,14]. Importantly, this illusion can induce motor [10,14], physical [11], physiological [15], cognitive and neural benefits [12,13] that are comparable to the effects that follow physical exercise. IVR is currently the only method that allows for the manipulation of many different parameters (visual, tactile, somatosensory, etc.) and can reproduce a full body, thereby inducing reactions on the real body at different levels. At the same time, the virtual illusion is so effective that actual movement is not required to produce the benefits that are typically associated, for example, with physical exercise [10,11,12].

Starting from the evidence that psychosocial stress reduces after physical activity and capitalizing on the above-mentioned opportunities that are offered by IVR, we investigated the psychosocial stress response of subjects after virtual training was performed exclusively by their own virtual body (while the person was not performing any actual movement) compared to virtual training that was performed by another virtual body. We expected to find a significant decrease in the psychosocial stress response, as measured by physiological measurements (salivary alpha-amylase, heart rate, etc.) and psychological self-reports (state anxiety), after training with the subject’s own avatar and not after training that was performed by another virtual body.

## 2. Materials and Methods

The protocol of the present randomized controlled trial (RCT) was developed in accordance with the CONSORT guidelines (Appendix A). The study was conducted in accordance with the Declaration of Helsinki and it was approved by the ethical committee of the Tohoku University Graduate School of Medicine (protocol number 2020-1-417) on 30 July 2020 and was registered to the University Hospital Medical Information Network (UMIN000041124) on 16 July 2020.

### 2.1. Participants

Volunteers for this study were recruited via the internet in the Tohoku region (Japan). They were all Japanese citizens, native Japanese speakers and university students. Our exclusion criteria comprised: self-reported current or past neurological, psychiatric (especially in the mood domain) or motor disorders; heavy alcohol consumers (more than 15 units a week for males and 8 for females) and avid smokers (more than 25 cigarettes per day); people currently using psychotropic drugs; people working night shifts [16]; women undergoing hormonal contraception therapies; and women who, at the time of the experimental sessions, were not in the luteal phase of their menstrual cycle [17]. We also asked individuals who were easily prone to motion sickness and people with previous direct experience of IVR to refrain from participating. Volunteers were compensated for their participation.

The sample size was estimated using G*Power 3.1 (Universitat Kiel, Kiel, Germany) with an a priori power analysis for an F test with within–between interaction. Considering the objective measurement for stress response (i.e., the salivary alpha-amylase) as the main outcome, we set a moderate effect size (f(V) = 0.4) [18,19] and calculated a total sample size of 52 participants (with the α error probability set at 0.05 and the power set at 0.80).

In total, 52 healthy young adults (24 females; average age: 22.01 ± 0.24 years; average education: 16.05 ± 0.21 years) entered and completed the entire RCT and were included in the final analysis (see Appendix A). After enrollment, participants were allocated into one of two study arms: a first person perspective group (hereinafter referred to as 1PP group: N = 26), which was the experimental group, and a third person perspective group (hereinafter referred to as 3PP group: N = 26), which was the control group [10,14]. Group assignment was managed by a co-experimenter, who did not take part in data collection, using a simple sequence of a 1:1 (experimental:control) ratio. The allocation of participants into each arm was based on the order of entry into the study, thereby ensuring the equivalence of the groups in terms of gender distribution [20]. Participants were fully aware of their group allocation because they had to experience the IVR under specific conditions, which rendered full blinding unfeasible. Appendix A summarizes the descriptive statistics of the entire sample and the group distribution (none of statistics were significantly different between the groups, which meant that the participants were homogeneously distributed and that the two groups were comparable).

### 2.2. Procedure

This RCT was organized into two sessions on two different dates, which were at the same time exactly ten weeks apart, to avoid any problems that were related to the repetition of the psychosocial stressor [21,22] (see Section 2.2.1 for details). All data were collected between 12 pm and 6 pm to control for diurnal variations [16].

For both sessions, participants were asked to refrain from eating, drinking anything other than water and smoking for at least 2 h before coming to the laboratory in order to avoid any intake that might interfere with their stress response (such as caffeine) and to avoid any external contamination of their salivary samples [22].

At the beginning of the first session, participants read an information sheet and signed a consent form. Then, they completed several self-administered surveys regarding some general information (smoking, hours of sleep of the previous night, ongoing pharmacological treatments and phase of menstruation cycle for women) and some important aspects concerning our experimental hypothesis: the Hospital Anxiety and Depression Scale (HADS), which includes two subscales for anxiety and depression [23]; the Perceived Stress Scale (PSS) to evaluate the frequency of stressful feelings and behaviors [24]; the trait subscale of the State–Trait Anxiety Inventory (STAI-T) was used to measure trait anxiety levels [25]; the International Physical Activity Questionnaire (IPAQ) to assess general engagement in physical activity and exercise since the experiment could have an effect on heart rate, especially during virtual training; and lastly, the Edinburgh Handedness Inventory (EHI), which concerns manual laterality.

Subsequently, the participants then underwent the Trier social stress test (TSST), which was used as the baseline assessment (see Section 2.2.1).

During the second session, the participants repeated the TSST, but this time following an immersive virtual reality training (IVRt) session under one of two different conditions, according to their group allocation (1PP or 3PP: see Section 2.2.2).

At specific time points (*t*) during both TSSTs, we collected salivary samples to measure alpha-amylase levels as a biomarker for the psychosocial stress response (main outcome) and state anxiety level. In addition, we continuously recorded their heart rate. We also asked the participants to answer an online (while completing the IVRt) and offline (immediately after the IVRt) questionnaire about their sense of ownership and agency.

#### 2.2.1. Trier Social Stress Test (TSST)

The Trier social stress test (TSST) is one of the most commonly used standardized behavioral tests to induce and evaluate psychosocial stress response because it combines high levels of social evaluative threat and uncontrollability [16,26,27].

The procedure that was used in this study was the same as that in [26], including the speech preparation (10 min), speech delivery (5 min), mathematical task (5 min) and resting phases (40 min, divided into two phases of 20 min each). To directly compare the effectiveness of the stress response to the IVRt, the TSST was delivered before (TSST baseline) and after the virtual training session (TSST post-training), i.e., the first and second experimental sessions, respectively. To limit the number of repetitions of the TSST (which may cause learning and habituation effects), we organized the study in a between-groups design.

The general procedure of the TSST was the same for both sessions. The setting was organized into two wide and quiet rooms: one for the speech preparation, delivery and math task phases and the other for the resting phase. To begin the test, the participant, who was wearing a heart rate recorder, sat on one side of a table in front of two experimenters (a man and a woman, as in [21]). A video camera was also placed right in front of the participant to record the whole session. One of the experimenters read the instructions for the first part of the test, which was framed as a mock job interview for the participant’s ideal job. The experimenter explained that they had to mentally prepare a 5-min speech, which would be evaluated by a panel of judges who were trained in public speaking. The participant was given 10 min to prepare on their own and both experimenters left the room (speech preparation phase). At the end of the preparation time, the experimenters re-entered the room wearing lab coats and sat back in their original place. After starting the camera recording, the same experimenter gave the signal for the participant to speak for 5 min, making sure to always look at the camera (speech delivery phase). While the participant was speaking, both experimenters maintained eye contact with the subject and refrained from making emotional facial expressions. If the participant stopped talking during the speech, the experimenter allowed them to remain silent for 20 s before reminding them that there still was time left. After 5 min, the experimenter stopped the participant and proceeded to the next step.

In the mathematical task, the participant was asked to sequentially subtract the number 13 from 1022 and report their answers aloud (mathematical task phase). After every mistake, the experimenter asked the participant to start the task over. This phase lasted for 5 min as well. Finally, the participant was invited to relax (but not sleep) for 40 min in a different room that contained comfortable sofas and neutral reading material (resting phase). The speech delivery phase and the mathematical task are commonly considered to be the active phases of the TSST, during which the stress response should reach its peak [26,28,29]. At the end of the entire RCT procedure, the participants were debriefed as to the true nature of the experiment and they were informed that their performance was actually recorded but then immediately deleted and that no analysis of their performance would be conducted.

In the second session, we used the same general setting and phases, but we introduced the following differences to avoid habituation, learning or repetition effects on the TSST [16,22,29,30,31,32]. First, the second session was administered 10 weeks after the first session, but we kept the timing constant for each participant (for example, a person who attended the first session on 4 May at 2 p.m. then attended the second on 13 July at 2 p.m.). A 10-week interval has been described as challenging enough for HPA axis reactivity to induce a significant stress level and ensure comparable results to those from previous repetitions of the TSST [21,22]. In order to ensure that the measurements that were taken during the first session were comparable to those that were right before the second session (even though they were collected 10 weeks later, they were still considered a baseline assessment), we also repeated and compared the pre-stress measurements immediately before the repetition of the TSST.

Secondly, we used different rooms to avoid familiarity with the environment, which might alter the stress response (as in [16]). Thirdly, the roles of the experimenters were exchanged: the experimenter who spoke during the first session remained silent during the second session, and vice versa, to avoid acquaintance with one of the researchers and alterations to the stress response. As mentioned before, the experimenters were a men and a woman because the sex of the interacting researcher can affect the stress response of the participant [16,33].

Lastly, we slightly modified the details of both the speech and mathematical tasks. Regarding the speech task, a recent meta-analytic review [34] identified novelty, lack of control, unpredictability and social evaluative threat as primary factors in acute psychosocial stressors. Previous studies have used different strategies to modify the speech content between the first and second repetitions (in addition to the time interval) but maintained those factors, such as describing the participant’s own positive/negative aspects [21] or speaking about a given topic [35]. Initially, we planned to use the same strategy, but since practically all participants essentially focused on their own presentation during the mock job interview (which already included details of their characteristics), we needed to modify the task. Therefore, we asked participants to present and publicize their hometown (to keep the individual element and not deviate excessively from the topic of the first session, which might affect the stress response) to a panel of experts on tourism marketing (as social evaluation). To maintain uncontrollability and unpredictability, during the first session, the participants were not informed about the content of the second session’s speech. In the mathematical task, the participants had to sequentially subtract the number 17 from 1035 to avoid learning and repetition effects [21,22].

##### Measurements during the TSST: sAA and STAI-S

In order to evaluate the neuroendocrine stress level of the participants during the TSST, we collected salivary samples using commercially available disposable strips (Salivary Amylase Monitor Strips, NIPRO Corporation, Osaka, Japan) to assess their level of salivary alpha-amylase (sAA) as a proxy measurement for the response of the autonomic nervous system to acute stress [5].

Although sAA levels are traditionally reported in combination with cortisol, sAA has also been reported as a reliable outcome measure by itself in several studies [36,37,38,39,40]. Indeed, the intensity of sAA and cortisol release has been found to be strongly interconnected during a stressor event [28], thereby suggesting their comparability. Additionally, in terms of timing, sAA release seems to be more immediate than that of cortisol [5] and even easier to activate during low- or mild-intensity stressors [41]. Consequently, sAA was more suitable for the purposes of this study.

At first, we asked the participants to rinse their mouths with water. They had to insert the tip of the strip (where a specific nonwoven material tape was placed by the manufacturer to absorb saliva) under the tongue for 30 s. After that, the strip was inserted into the monitor device where it came into contact with the reagents that were already added by the manufacturer and then, the transcript product was optically analyzed. The sAA level (the main outcome of this study) was obtained and noted. Then, the strip was immediately discarded and no biological samples were stored. The sAA level was expressed in kU/L.

As a subjective counterpart, the state subscale of the State–Trait Anxiety Inventory (STAI-S) was used [25], which includes 20 self-administered statements about current levels of anxiety using a 1–4 Likert scale, where 1 means “not at all” and 4 means “completely”. Total scores ranged from 20 to 80 (higher scores meant higher levels of state anxiety).

The results of both the sAA and STAI-S were collected at the same time points throughout the RCT: the very beginning of the pre-stress phase (*t*_1_); immediately after the mathematical task of the TSST (*t*_4_), which should be the peak moment for the stress response [26]; and 20 (*t*_5_) and 40 mins (*t*_6_) into the resting phase. In addition, for the second session only, they were also recorded immediately after the IVR training (*t*_0_) to assess differences in stress response that were due to the virtual training (see Figure 1).

Heart rate (HR) was recorded using a Polar H10 (Polar Electro, Kempele, Finland) chest monitor, which was connected to a smartphone via Bluetooth on which a specific application recorded all data (the instantaneous heart rate, expressed in beats per minute (bpm; as in [11,12,13]), changes once per second). HR was recorded at rest for 5 min before the start of the TSST in the first session, before the beginning of the IVR training in the second session (pre-stress phases) and continuously throughout each phase of the TSST, excluding the resting phase, as in [26].

#### 2.2.2. Immersive Virtual Reality Training (IVRt)

Before the participants wore the IVR visor, the experimenter explained how the training and the questionnaire were to be administered. In particular, the experimenter explained the correct position to maintain: participants had to sit on a footstool with their feet firmly touching the ground and their arms relaxed along the sides of their body (Figure 2A,D). Participants were instructed not to move their bodies throughout the whole experience; however, they could rotate their heads and the virtual scene would simultaneously and coherently change to maintain their perspective as realistically and plausibly as in real life. The participants wore the Oculus Rift visor (oculus.com, accessed on 6 February 2020), which has two lenses that are positioned in correspondence to the user’s eyes, and were shown a virtual environment (modeled in Unity3D), which was composed of a simple open space with a meadow-like green floor and a natural-like bright sky that were separated by a visible horizon. The avatars that were used in this study were gender-matched and life-sized humanoid standing bodies, which were downloaded from the public Microsoft Rocketbox Avatar library [42]. Participants were instructed to always look at the virtual body, especially the legs. For the entire duration of the IVRt, we repeatedly asked the participants to report any feelings of discomfort, nausea, sickness, etc.

The immersive virtual reality training (IVRt) included two phases: during the first 3 mins (static phase), the avatar stayed still, thereby allowing the participants to familiarize themselves with the virtual setting and to control for potential dizziness or sickness; for the following 30 mins (dynamic phase), the avatar ran at a constant speed of 6.4 km/h, which is considered to be moderate-intensity exercise for young people with moderate fitness levels [17,18].

The avatar was displayed from the first person perspective (1PP) or the third person perspective (3PP), according to the group allocation of each subject (Figure 2). For 1PP (the experimental group), the virtual body substituted and spatially replaced the real body of the participant. For 3PP (the control group), the same virtual body was displayed around 1.5 m to the left of the real participant’s body, as in [10,11,13].

##### Measurements during the IVRt: HR and Online and Offline Questionnaires

During the IVRt, we exploited three measurements: heart rate, an online questionnaire regarding the sense of body ownership and agency during the IVRt itself and an offline questionnaire to explore other aspects of the subjective experience of the IVR in more detail [10,12,13].

Heart rate was recorded continuously during the static and dynamic phases of the IVRt using the same equipment that was used for the recording of HR during the TSST (see “Measurements during the TSST: sAA and STAI-S” part in Section 2.2.1).

The online questionnaire consisted of four statements (from s1 to s4) that were read aloud by the experimenter in a random order, which participants had to answer orally using a 1 (“I strongly disagree”) to 7 (“I strongly agree”) Likert scale. The same four statements were repeated four times in total to account for possible differences in the strength of the illusion [10,12]: at 1:30 min into the static phase and at minutes 5, 15 and 25 of the dynamic phase.

At the end of the IVRt, the participants took off the visor and completed the self-report offline questionnaire (from s5 to s15) using the same Likert scale that was mentioned before. See Table 1 for all of the statements.

### 2.3. Statistical Analysis

All values that are displayed are the average ± standard error (SE). The significance level was set at *p* < 0.05. The distribution was assessed using a Shapiro–Wilk test. When the data were normally distributed, a repeated measure ANOVA was used for between–within group comparisons and post hoc analyses were conducted with Duncan correction. For the latter, the effect sizes were expressed as partial eta squared (η_p_^2^: 0.01 = small, 0.06 = medium, 0.14 = large effect size) [19] and observed power (op) was reported when necessary. The ά error probability was always set at 0.05. When the data were not normally distributed, the Mann–Whitney U test was performed for between-groups comparisons and the Wilcoxon matched-pair test was used for within-groups comparisons. For significant results, the effect sizes were reported using the coefficient of determination (R^2:^ < 0.13 = small, 0.13 < R^2^ > 0.26 = medium, >0.26 = large effect size) [43].

#### 2.3.1. Online and Offline Questionnaires Regarding the IVRt

The raw data underwent an intra-individual ipsatization procedure to obtain their z-scores, as in [44,45]. For the online questionnaire, once it was confirmed that there were no differences between the repetitions using Friedman ANOVA and considering each statement separately with its repetitions, independently from group assignment (for example, s1 at 5 min, s1 at 15 min, s1 at 25 min), the data from each statement were averaged and considered independently from the others. The first repetition of the statements, i.e., during the static phase, was also considered separately from the three repetitions during the dynamic phase since it was intrinsically different and only the first two statements were included for the static phase because the avatar had not moved yet, so there was no sense of agency involved.

Since all of the data were non-normally distributed, we ran a Mann–Whitney U test to compare the 1PP and 3PP groups.

#### 2.3.2. HR during the IVRt

The HR data from each phase were averaged for each subject and analyzed by means of a 2 × 2 repeated measure ANOVA, using PHASE (static and dynamic) as the within factor and GROUP (1PP and 3PP) as the between factor. To account for the general physical activity level, we also used the IPAQ test as a covariate by means of a ranked ANCOVA (Quade’s test) with Duncan correction.

#### 2.3.3. sAA during the TSST

First, for each subject, we subtracted their data from the first session from their data from the second session (TSST post-training − TSST baseline) in order to obtain the clean effects of the intervention itself at each of the time points (t) of the evaluation and the delta time points (dt). Then, due to the non-normal distribution of the sAA data, we used Mann–Whitney U test analyses with TIMEPOINTS as the within variable (dt_1_, dt_4_, dt_5_ and dt_6_) and GROUP (1PP and 3PP) as the between factor. Additional comparisons within groups and/or within sessions were then run using a Wilcoxon matched-pair test. Particular emphasis was allocated to dt_4_ and t_4_ (immediately after the math task of the TSST), which was supposed to be the peak for stress response.

Additionally, we also compared the sAA levels of the groups to control for the results of the HADS (anxiety and depression subscales), PSS and STAI-T, as covariates, by means of ranked ANCOVAs (Quade’s tests) with Duncan correction, since general levels of stress and anxiety can affect the release of stress-related enzymes.

#### 2.3.4. STAI-S during the TSST

The scores for each STAI-S questionnaire were first calculated. Then, as for the sAA data, we subtracted the data that were collected from the STAI-S in the first session from the data from the second session (TSST post-training − TSST baseline) so that we could obtain the same delta time points. The STAI-S data were normally distributed, so we ran a 4 × 2 repeated measure ANOVA with TIMEPOINTS as the within variable (dt_1_, dt_4_, dt_5_ and dt_6_) and GROUP as the between factor (1PP and 3PP).

We also compared the STAI-S results of the groups to control for the results of the STAI-T, as a covariate, by means of ANCOVA with Duncan correction.

#### 2.3.5. HR during the TSST

As HR was recorded continuously throughout the entire duration of each phase of the TSST, we averaged the data for each subject and each phase of the TSST to obtain t_1_ (the 5-min recording from the pre-stress phase), t_2_ (the 10-min recording from the speech preparation phase), t_3_ (the 5-min recording from the speech delivery phase) and t_4_ (the 5-min recording from the math task) for each session. As for the sAA data, we first subtracted the data from Session 1 from the data from Session 2 (TSST post-training − TSST baseline) to obtain the delta time points (dt_1_, dt_2_, dt_3_ and dt_4_). Since the HR data were normally distributed, we ran a 4 × 2 ANOVA with TIMEPOINT as the within factor and GROUP (1PP and 3PP) as the between factor.

#### 2.3.6. Correlations

The measurements from during the TSST (sAA, STAI-S and HR) and those from during the IVRt (online and offline questionnaires and HR) were tested for correlations: first within each other (for example, correlations between the sAA and STAI-S during the TSST or between the online questionnaire and HR during the IVRt) and then between each other (for example, correlations between the online questionnaire and sAA). Particularly important were correlations between objective and subjective measurements. According to the data distribution, either Spearman’s or Pearson’s correlations were used.

## 3. Results

### 3.1. Surveys

For the surveys regarding demographic and general information, the average score for the anxiety and depression subscales of the HADS for the whole sample was below 7, which is considered “normal”. One subject was an outlier (from the 3PP group), showing a score of 13 for each of the subscales (considered “moderate”). Regarding the PSS, scores between 14 and 26 are interpreted as “moderate stress”, as in the case of the present sample [24]. Additionally, for the STAI-T, a cut-off of 39–40 is suggested to detect significant anxiety symptoms (with higher scores indicating greater anxiety). In the present case, a score of 41 referred to moderate state anxiety [25]. Finally, for the survey regarding physical activity (the IPAQ), the vast majority of the sample reported a “moderate” level of physical activity. Combined with a normal body mass index, the participants were considered to be physically healthy (see Appendix A).

### 3.2. Measurements during the IVRt

#### 3.2.1. Online Questionnaire

All data (z-score and data from the 1–7 Likert scale) are reported in Table 1. During the static phase, the U test to compare the groups showed that the results for s1 (sense of body ownership) were significantly different (*p* < 0.01, 2*1-sided exact *p* < 0.01, adjusted z = 5.93, R^2^ = 0.67) between the 1PP group and the 3PP group, with the results of the 1PP group being higher than those of the control group. As for s2 (the control statement for the sense of body ownership), we found the exact opposite pattern: in 1PP, the results for s2 were significantly lower (*p* < *0*.01, 2*1-sided exact *p* < *0*.01, adjusted z = −5.87, R^2^ = 0.66) than those of 3PP.

During the dynamic phase, the answer patterns for sense of body ownership that were observed during the static phase seemed to be maintained. For the sense of body ownership, the results for s1 were significantly higher (*p* < 0.01, 2*1-sided exact *p* < 0.01, adjusted z = 5.47, R^2^ = 0.57) in the 1PP group than those in the 3PP group, while the results for s2 (control statement) were significantly lower (*p* < 0.01, 2*1-sided exact *p* < 0.01, adjusted z = −4.26, R^2^ = 0.34) in the 1PP group than those in the 3PP group. Additionally, for the sense of agency, the result for s3 were significantly different (*p* < 0.01, 2*1-sided exact *p* < 0.01, adjusted z = 4.07, R^2^ = 0.42), with the answers from the 1PP group being higher than those in the 3PP group. The results for the control statement for agency (s4) did not differ between the groups.

#### 3.2.2. Offline Questionnaire

According to the U test, several statements in the online questionnaire were reported as being significantly different between the groups, with the scores of the 1PP group being higher than those of 3PP (see all z-scores and raw data in Table 1). The results for s5, which was about sense of spatial location (*p* = 0.04, 2*1-sided exact *p* < 0.04, adjusted z = 2.03; R^2^ = 0.07), s6, which was about sense of body ownership (*p* = 0.01, 2*1-sided exact *p* = 0.01, adjusted z = 2.67, R^2^ = 0.13), s7, which was about the feeling of standing (*p* < 0.01, 2*1-sided exact *p* < 0.01, adjusted z = 3.11, R^2^ = 0.18), s8, which concerned ownership of movement (*p* < 0.01, 2*1-sided exact *p* < 0.01, adjusted z = 3.23, R^2^ = 0.20), s9, which was about sense of agency (*p* < 0.01, 2*1-sided exact *p* < 0.01, adjusted z = 3.05, R^2^ = 0.17), s11 (about sense of effort), s12 (about vection) and s13 (about the feeling of walking) were all not significantly different.

#### 3.2.3. HR during the IVRt

The 2 × 2 ANOVA to compare the HR data during the IVRt showed a significant interaction (F(1, 50) = 5.75, *p* = 0.02, η_p_^2^ = 0.10, op = 0.65), a significant main effect of GROUP (F(1, 50) = 5.93, *p* = 0.02, η_p_^2^ = 0.10, op = 0.66) and a significant main effect of PHASE (F(1, 50) = 6.43, *p* = 0.01, η_p_^2^ = 0.11, op = 0.70). The average HR during the dynamic phase was significantly higher in the 1PP group (87.26 ± 2.10) than that in 3PP (77.93 ± 1.92) and with respect to the static phase for both groups (1PP: 81.68 ± 2.44; 3PP: 77.78 ± 1.78) (*p* < 0.01 for all post hoc comparisons). For the static phase, the comparison between the groups was not significant.

Even when controlling for the IPAQ kcal results, as a covariate, the main GROUP factor was still significant according to the ranked ANCOVA (F(1, 50) = 9.34, *p* < 0.01, η_p_^2^ = 0.15, op = 0.85).

#### 3.2.4. Correlations between Measurements during the IVRt

At first, we checked for correlations between the measurements from during the IVRt. Using Spearman’s correlation, the HR during the IVRt was positively correlated with s1 (r = 0.46, *p* < 0.01) and s3 (r = 0.27, *p* = 0.04) from the online questionnaire, meaning that HR increased coherently with the sense of body ownership and agency. As for the offline questionnaire, we found negative correlations between HR and s10, which was a control statement for the sense of body ownership (r = −0.34, *p* < 0.01), and s14, which was about the feeling of being dragged (r = −0.31, *p* = 0.03), meaning that HR decreased when the avatar was perceived more as belonging to someone else and when the subjects experienced the feeling of being dragged. Additionally, we correlated the answers to the online questionnaire with those to the offline questionnaire (Appendix A).

It is worth noting that s1 and s2, which were about online sense of body ownership and agency, respectively, positively correlated with s7, which was about offline sense of standing (s1: r = 0.37, *p* = 0.01; s2: r = 0.32, *p* = 0.04), s8, which was about offline sense of agency (s1: r = 0.45, *p* = 0.01; s2: r = 0.44, *p* = 0.01), and s9, which was also about agency (s1: r = 0.42, *p* = 0.01; s2: r = 0.46, *p* = 0.01). Both s1 and s3, also negatively correlated with s10, which was the control statement for offline sense of ownership (s1: r = −0.56, *p* = 0.01; s2: r = −0.56, *p* = 0.01), and s15, which was about the feeling of sliding (s1: r = −0.43, *p* = 0.01; s2: r = −0.30, *p* = 0.04). Additionally, s3 also positively correlated with s11, which was about sense of effort (r = 0.35, *p* = 0.01), and s13, which was about the feeling of walking (r = 0.30, *p* = 0.04). In contrast, s2, which was the control statement for the online sense of ownership, mostly negatively correlated with the offline statements, except for s10 (which was also a control statement for offline sense of ownership) and s15 (which was about the feeling of sliding).

### 3.3. Measurements during the TSST

#### 3.3.1. sAA

Table 2 reports the descriptive statistics for the sAA measurements.

According to the U test, dt_4_ (immediately after the math task of the TSST, i.e., the peak of the active stress response) was significantly different (*p* < 0.01, 2*1-sided exact *p* < 0.01, adjusted z = −3.87, R^2^ = 1.62) between the groups, with lower sAA levels in the 1PP group (−61.69 ± 9.13) compared to the 3PP group (−11.23 ± 5.14). Additionally, dt_5_ also showed the same pattern, with the results of the 1PP group (−52.38 ± 7.62) being significantly lower (*p* < 0.01, 2*1-sided exact *p* < 0.01, adjusted z = −2.74, R^2^ = 0.98) than those of the 3PP group (−19.03 ± 10.06). For PP1 (−11.88 ± 5.56), dt1 was also significantly lower (*p* = 0.02, 2*1-sided exact *p* = 0.02, adjusted z = −2.39, R^2^ = 0.54) than for 3PP (10.38 ± 6.04). However, dt_6_ did not differ between the groups (see Figure 3).

To test for the effects of the virtual training itself on stress levels, sAA was also measured immediately after the IVRt (t0) in the second session. Only considering the sAA variable at t0, we found no significant differences between the groups, even though 1PP (114.11 ± 17.44) showed higher sAA values than 3PP (88.26 ± 16.40).

According to the ranked ANCOVAs, the main GROUP factor remained significant when controlling for the anxiety subscale of the HADS results (F(1, 50) = 15.45, *p* < *0*.01, η_p_^2^= 0.23, op = 0.97), the depression subscale of the HADS (F(1, 50) = 15.59, *p* < 0.01, η_p_^2^= 0.23, op = 0.97), the PSS (F(1, 50) = 15.17, *p* < 0.01, η_p_^2^= 0.23, op = 0.96) and the STAI-T results (F(1, 50) = 14.26, *p* < 0.01, η_p_^2^= 0.22, op = 0.96). According to the post hoc comparisons, the same differences were maintained when controlling for the above-mentioned covariates.

#### 3.3.2. STAI-S

Table 3 reports the descriptive statistics for the STAI-S measurements.

According to the 4 × 2 ANOVA, the interaction between TIMEPOINTS and GROUP was significant (F(3, 150) = 2.92, *p* = 0.04, η_p_^2^ = 0.05, op = 0.68). According to the post hoc comparisons, the dt_4_ of the 1PP group (−6.76 ± 2.04) was significantly lower than all other measurements (*p* < 0.05 for all comparisons). None of the other comparisons were significant. In particular, the first measurement (dt_1_) was not significantly different between the groups (see Figure 4).

We also analyzed the results from Session 1 separately to check whether, from a subjective perspective, the TSST had an effect on the sample before the introduction of the virtual illusion. Considering all participants together and according to the repeated measures ANOVA with TIMEPOINT (four levels) as the within factor, we found that the main effect was significant (F(3, 153) = 61.07, *p* < 0.01, η_p_^2^ = 0.54, op = 0.99), with the post hoc comparisons showing that t_4_ (46.15 ± 1.19) was significantly higher than all other time points (*p* < 0.01 for all comparisons). Considering the two groups separately for Session 1, the GROUP factor was not significant, meaning that they all experienced stress during the TSST, independent from group allocation.

As for the sAA measurements, we also measured the STAI-S during the second session immediately after the IVRt. According to the two-tailed *t* test, we found that the scores after the IVRt were not different between the groups. As for the sAA results, we can confirm that we found no differences between the t_1_ values of the two sessions, even considering the two groups separately.

However, when we controlled for the STAI-T in the ANCOVA, the main GROUP effect was slightly above the significance level (*p* = 0.06).

#### 3.3.3. HR during TSST

Table 4 reports the descriptive statistics for the HR measurements during the TSST.

Even though the 1PP group showed lower HR values than the 3PP group (in particular, dt_2_, dt_3_ and dt_4_ in 1PP were the lowest values), the 4 × 2 ANOVA did not find any significant effects of GROUP or TIMEPOINT (see Figure 5).

When we examined the results from all participants (not divided by group) during Session 1 only, we found a significant main effect of TIMEPOINT (F(3, 153) = 35.9, *p* < 0.01, η_p_^2^ = 0.41, op = 0.99), where t_3_ was significantly higher than all other measurements (*p* < 0.01 for all comparisons). When we considered the group allocation, the main effect of GROUP was not significant (the two patterns practically overlap), meaning that all participants experienced an increased stress response during the TSST, especially after the preparation, speech and math task phases. Additionally, the two groups were comparable.

#### 3.3.4. Correlations between Measurements from the TSST and IVRt

Then, we checked for correlations between all measurements during the TSST. As for the sAA levels (the main outcome of the study), we did not find any significant correlations with the HR and STAI-S measurements. Despite this, we found a significant negative correlation between the HR and STAI-S results at dt_4_ (r = −0.27, *p* = 0.04).

Lastly, we correlated the data from the IVRt with those from the TSST. Interestingly, we found a significant negative correlation between HR during the IVRt and the sAA level at dt_2_ (r = −0.31, *p* = 0.02), which suggests that the more HR increased during the virtual training, the less stress was measured by sAA levels. Coherently, we found a negative correlation between s1 from the online questionnaire and sAA level at dt2 (r = −0.35, *p* = 0.01) but a positive correlation with s2 (r = 0.29, *p* = 0.03) and a positive correlation between s10 and sAA at the same time point (r = 0.36, *p* = 0.01), suggesting that the more body ownership that was experienced during and after the IVRt, the less stress was measured right after the TSST. Additionally, the HR measurements from during the IVRt and during the TSST negatively correlate at all time points (see Appendix A).

## 4. Discussion

### 4.1. For Novelty, We Demonstrated That Training Performed by the Subject’s Own Virtual Body Could Induce Neuroendocrine Changes

It has been described in many studies that physical activity can have a direct effect on stress reduction [3], mostly as defined by the cross-stressor adaptation hypothesis [4]. However, to the best of our knowledge, this is the first attempt to show comparable effects from training that is performed exclusively by the subject’s own virtual body. While the baseline TSST results showed the typical pattern of stress response in terms of sAA levels, the most important aspect was the difference between the post-training results of the two groups, particularly regarding the measurement that was taken right after the active phase of stress induction (i.e., after the math task). We found that only those who experienced the IVRt from 1PP reported lower levels of sAA.

Even compared to acute exercise, we found similar results to when the training was physically performed [18], even though the comparison between real and virtual exercise was not a goal of this study. This is in line with previous studies, which found that specific virtual training was able to induce acute or short-term beneficial effects on parameters that were different from the mere perceptual level [10,12,13,46,47,48]; for example, in [47,49], the link between the illusory perception of the subject’s own body and higher cognitive functions was interpreted in terms of the redefinition of self via the perspective taking process. A person embodying a transformed body redefines themselves based on their new perceptual characteristics, thereby providing the possibility to access mental resources that are normally limited by their usual way of thinking about themselves. In our previous study [12], we argued that the illusory sense of ownership and agency over an avatar is strong enough to initiate a cascade of events that is comparable to those that occur in the physical world. Specifically, the manipulation of bodily perception leads to a retrospective reappraisal of the sense of agency (i.e., a sense of body ownership leads to the sense of agency, which, in turn, leads to a physiological reaction that activates higher functions).

From the present results, we believe that we added a crucial contribution to the previous studies since the stress reduction was measured at the endocrine level (i.e., sAA levels). This novelty can be explained by both of the previously mentioned interpretations: the speculation about the “redefinition of the self” could be increased in light of the present results by including not only the newly transformed bodily self, but also the “moving bodily self”, meaning that the dynamic features of the embodied avatar can contribute to the novel available resources [47]. Our previous speculation can also be extended by including a bio-physiological level to the cascade of consequences, which occurs in parallel to the cognitive and neural effects [12].

Additionally, a parallel result was found for state anxiety. Here, the subjective level of state anxiety increased after the active phase of the TSST, as expected, but a reduced stress response emerged in those who experienced the IVRt from 1PP. These results reflect that stress is a multimodal concept comprising physiological and psychological processes [50]. The results are also of great relevance due to their potential clinical implications, especially considering that anxiety is one of the most frequently reported symptoms among the general population [51]. Notably, the levels of trait anxiety in our sample were relatively high, with STAI-T scores that were very close to the cut-off. The literature suggests that it is not uncommon to detect signs of psychosocial stress among healthy young adults, especially in countries with high performance demands, such as Japan [52]. Hence, the results of the present RCT can be generalized. On the other hand, it could be easier to induce the stress response (as we did using the TSST) in subjects with relatively high levels of trait anxiety and, therefore, the beneficial effects of the IVRt on state anxiety could have a very interesting impact on clinical research.

Lastly, we examined the results of the HR measurements. Even though during the TSST the results showed the typical pattern that has been described in previous studies with an average peak during the speech delivery phase, especially during the baseline TSST [16,21,26], there were no significant differences between the groups when observing the delta time points. The HR results from the 1PP group were slightly lower than those from the 3PP group due to the values in from the second session (after the IVRt), but the differences were still not significant. In previous studies, the possibility of having a discordance between subjective and objective measurements before and after the TSST has been reported. Patients who were diagnosed with social phobias showed higher levels of subjective anxiety both right after and one week after exposure to the stressor, but did not show any physiological changes (including HR changes) [53]. In the healthy population, a similar contrast has also been detected. A correlation between state and trait anxiety has been described, but not using objective anxiety-related measurements, including heart rate. The authors argued that participants with higher levels of trait anxiety tend to overestimate their current level of anxiety, which is not necessarily reflected on a behavioral level [54]. This could offer a partial explanation for the current results, since the recruited participants of this study also showed normal to high levels of trait anxiety. However, this effect did not extend to sAA levels, which were significantly different between the groups during the TSST post-training. Combining the above-mentioned explanations with the current results, we can conclude that in healthy people who show relatively high levels of trait anxiety, state anxiety is also high; however, this is not reflected in physiological measures (i.e., HR) but rather in endocrine measures (i.e., sAA levels), thereby acting on different levels.

### 4.2. Confirming Previous Results, Subjective Body Ownership and Agency Over a Moving Virtual Body Induced Physiological Changes

Overall, the results from the online questionnaire during the IVRt confirmed that the participants tended to subjectively perceive the virtual body and virtual movements as their own when displayed from 1PP but not when displayed from 3PP [9]. As previously described [11,12,13,14], the body ownership illusion is sustained, first of all, due to the bottom-up processes that are involved in the perspective. IVR provides the opportunity to completely replace and overlap the real with a virtual body, which induces a strong illusion of the sole visual stimulation that is different from that of other multisensory illusions that require additional stimulations (e.g., the rubber hand illusion [55]). Additionally, top-down processes provide an important contribution in terms of plausibility and realism [9,56].

While the ownership illusion was more established, the illusory sense of agency was feebler, according to the results for the statement about agency during the dynamic phase. Even though it was not a very strong effect, what made it remarkable was that there was a clear violation of the motor and intentional constraints that are typically associated with an illusory sense of agency, since the avatar was moving while the person was still. This was not the first case of the presence of an illusion under the conditions of a violation of temporal [57,58], motor or volitional [10,59] constraints, which interestingly also confirmed that the agency illusion could be sustained under certain circumstances. As argued in previous studies with similar effects [12,13], those circumstances could refer to (1) the presence of an agent with a strong sense of body ownership or (2) the absence of a specific motor intention because the person’s actual body was not visible, so the virtual (visible) body became the only possible agent with the necessary attributes for a sense of agency, which was even more plausible considering the strong sense of body ownership (during both static and dynamic phases). In addition, the participants here did not develop a specific motor intention since they did not have to actively perform any movements; therefore, it could be considered as an uncertain situation, which would explain the results of the questionnaire being slightly higher than “I don’t know”. Those two factors, together with the sense of body ownership, could explain the possibility of having a virtual illusion even under apparent conditions of a violation of intention and motor commands, thereby providing more relevance to body-related afferent signals and top-down mechanisms in the model of motor control [60].

The interpretation of the results from the online questionnaire could be extended to those from the offline questionnaire. Interestingly, we found that subjects in the 1PP group perceived themselves as standing, as when there is a sense of body ownership over an agent, it overwrites the contrary vestibular information. In fact, visual stimulation is usually considered to be predominant among body-related afferent signals, especially during movement.

The self-attribution of the seen movements (s8) could be explained by the interpretation of the movements per se: walking or running are considered to be automatic, cyclic and repetitive movements that are different from other goal-directed actions (such as reaching or kicking) and once the action is initiated, the very same movement pattern is then simply repeated [11]. This could be crucial in the creation of one of the most important efferent components of a sense of agency, i.e., prediction. Intuitively, a person can easily predict a movement that is repetitive rather than irregular. The second fundamental component of agency is intention: s9 referred directly to the intention of movement and was rated around 4 (“I don’t know”) in the 1PP group and around 2 (“I strongly disagree”) in the 3PP group. Perhaps the possibility of prediction that is afforded by cyclic action and uncertainty, in terms of motor intention, is enough to induce a judgment of agency [61]. Summarizing with the descriptions of afferent signals and sense of ownership that were mentioned previously, we argue that a retrospective sense of agency was created here (i.e., “this is my body, this body is moving, ergo these are my movements”) [12,62,63].

In line with the subjective data, the IVRt seemed to also be effective from an objective perspective, i.e., the heart rate data. Previous studies in which the avatar’s speed changed during the virtual training have shown that the participant’s HR adapts to the virtual movements, increasing or decreasing according to the speed or intensity of the virtual training despite the fact that the person is completely still [11,12,13]. However, in this case, the intensity of the virtual training was kept constant. The main factor seemed to be, once again, the ownership and agency illusion that was provided by the perspective. During the active phase, the average HR in 1PP was significantly higher than that in 3PP, meaning that when the moving virtual body was perceived as the subject’s own, the subject’s HR increased. Similar physiological effects have been observed in response to threats or dangerous stimuli that are directed at the considered-as-own body [64,65]. We believe that the physiological activation that was observed in the 1PP group was coherent with the illusory sense of ownership and agency that was induced by the virtual perspective (even though the HR values were not perfectly consistent with the actual speed of the virtual body), through which a physiological activation could also be induced as a consequence of the illusion itself, in addition to a reconstruction of the motor intention, which proceeded in a retrospective way [9,11,13,59].

Among the limitations of the present study, we could not entirely confirm the same effects as were found for the sAA levels of the subjective counterpart: the STAI-S. Even though anxiety levels actually diminished after the virtual training, there was not a significant effect when we accounted for trait anxiety. We argue that this could be due to the fact that the recruited participants showed trait anxiety levels that were close to the cut-off threshold for the general population. Future studies should consider the use of trait anxiety as an exclusion criterion or, alternatively, to select participants who have a diagnosis of mood or anxiety disorders. This study focused on the acute beneficial effects of virtual training and previous studies have shown that changing the target population (e.g., elderly instead young adults) and changing the training to a longer version (several sessions or weeks of training instead of one session only) can change the expected results [13]; therefore, we cannot declare whether the present results can be extended into the long-term. Future studies should include follow-up measurements and/or longer interventions. Regarding the sample, we only included young healthy individuals, but since the results of this study clearly have strong clinical implications, further research is warranted to test the efficacy of this training for clinical populations. Furthermore, even though we extensively described the reasons for the need to repeat the TSST, we could not control for a potential order effect, despite reduced stress only being detected after the IVRt among the 1PP group (confirming, at least partially, that the effect was not due to the order itself). Another possibility would be to use a different design, such as one subgroup performing TSST, IVRt and TSST and another subgroup performing IVRt, TSST and TSST. In this case, though, the baselines for the two subgroups would be performed at two very different moments (at least 10 weeks apart). It appears that sAA is influenced by the action of seasonality (mainly due to the changes in darkness/light) [66]; therefore, we opted for a more classical study design, but we could only partially control for the order effect.

Lastly and more generally speaking, even though we found effects in this study that could be approximately compared to those that occur after actual physical activity, it would be inappropriate and premature to suggest that IVR training could completely replace physical exercise. At the moment, the technological resources that are available pose some limitations and side effects (for example, difficulties in acceptance, especially by certain populations, and the absence of certain crucial components, such as musculoskeletal involvement and social aspects, as well as IVR-related sickness), just as with many other forms of intervention. Therefore, caution is recommended for the use of IVR training and further studies to explore the possible side effects of IVR training are necessary.

## 5. Conclusions

Along with beneficial effects for physiological [11], motor [14] and cognitive [12,13] functions, IVR training can also promote positive acute benefits at the neuroendocrine level (measured by sAA) and for subjective anxiety (measured by the STAI-S), thereby modulating the psychosocial stress response. Once again, we confirmed the strength of the virtual illusion of ownership and agency over the avatar, even under conditions of visuo-motor asynchrony and motor intention discrepancy.

## Figures and Tables

**Figure 1 ijerph-19-06340-f001:**
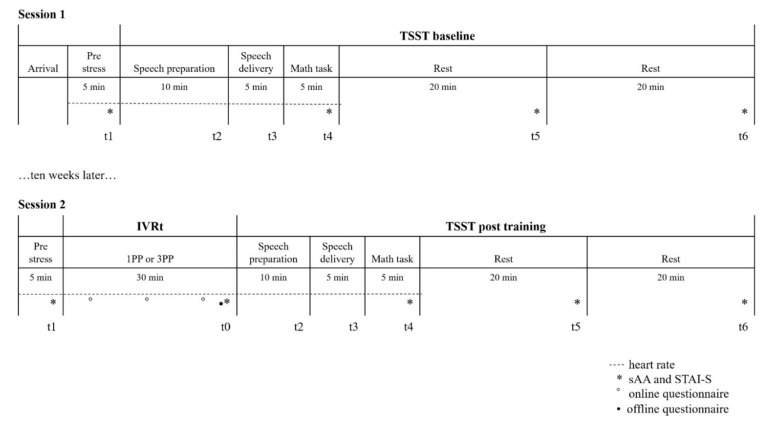
Schematic of the procedure in this study. The upper part of the figure represents the procedure for Session 1, including arrival (when participants completed the ethical documents and filled out the surveys), the pre-stress phase and the baseline TSST. The lower part represents the procedure for Session 2, in which, in addition to the same phases as Session 1, the virtual training (IVRt) was included. For each phase, the duration and corresponding time points were defined. The measurements are represented with symbols (as in the legend).

**Figure 2 ijerph-19-06340-f002:**
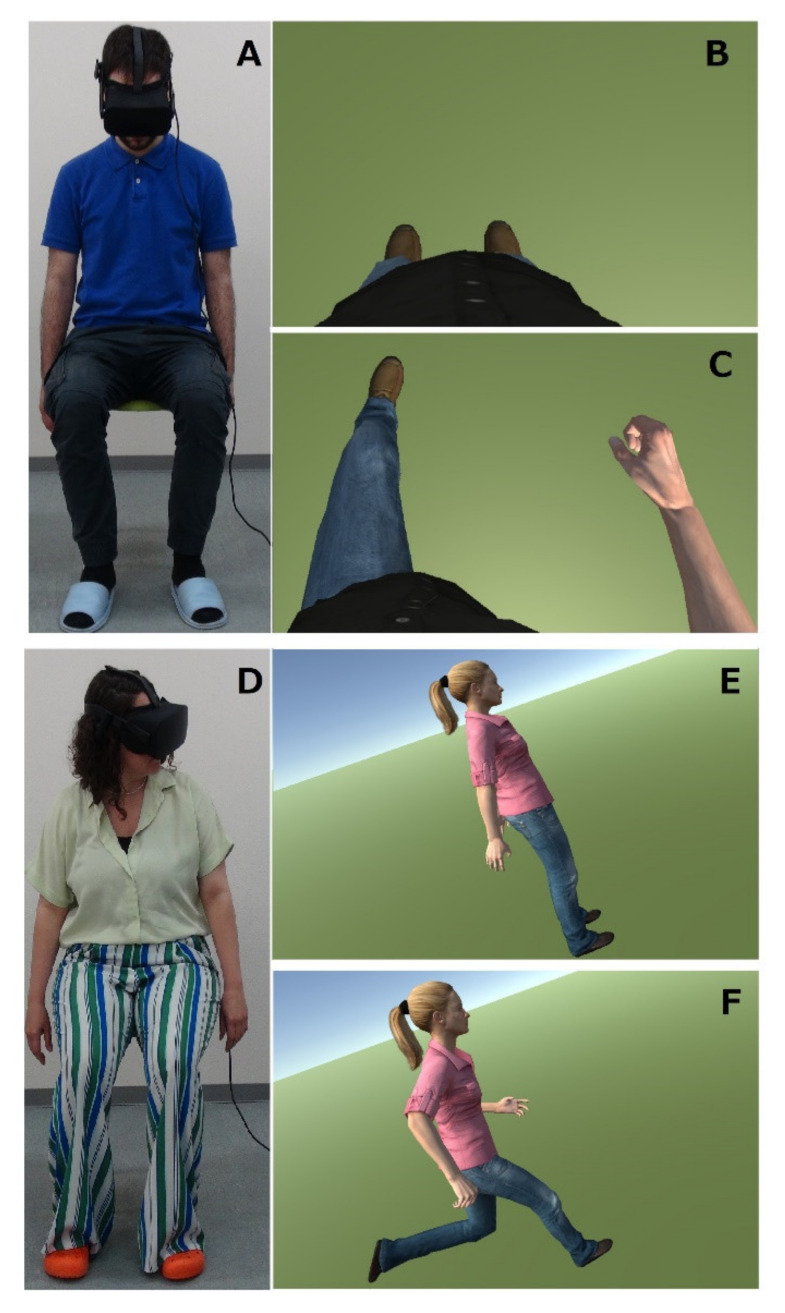
Participants and the virtual bodies in the virtual scenario: (**A**,**D**) male and female participants sitting, wearing the visor and looking in the direction that was coherent with their group assignment in order to see the avatar (the male is looking down, towards himself; the female is looking to her left side), respectively; (**B**) (static phase) and (**C**) (dynamic phase) the male virtual body displayed from 1PP; (**E**) (static phase) and (**F**) (dynamic phase) the female virtual body displayed from 3PP.

**Figure 3 ijerph-19-06340-f003:**
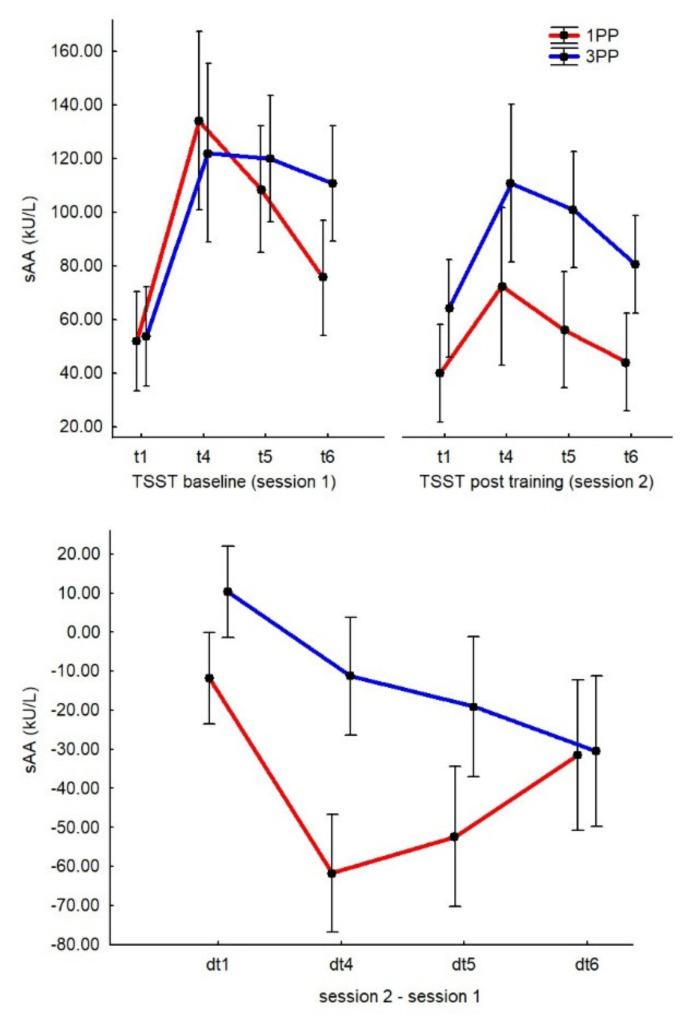
Graphs of the sAA results. The top chart represents the two sessions separately and the bottom chart represents the differences between sessions. In the top graph, time points (t) of the sAA measurements are represented as follows (also see Figure 1 as reference): the very beginning of the pre-stress phase (t1), immediately after the mathematical task of the TSST (t4) and after 20 (t5) and 40 min (t6) of the resting phase of the TSST. In the bottom graph, the delta time points (dt) represent the data from Session 2 minus the data from Session 1 (TSST post-training − TSST baseline, e.g., dt1 = t1 of Session 2−t1 of Session 1). Furthermore, we analyzed the data from Session 1 only (TSST baseline) to evaluate the effects of the TSST on the sample, independently from the IVRt. Considering all 52 subjects and according to the Wilcoxon-matched pair test, t4 (128.05 ± 11.64) was significantly higher (*p* < 0.01, z = 5.62, R^2^ = 0.35) than t1 (52.80 ± 6.47) and t6 (93.21 ± 7.89; *p* < 0.01, z = 3.21, R^2^ = 0.35), but not different from t5 (114.25 ± 8.24). Considering the two groups separately (using U tests), none of the time points were different, except t6 (*p* = 0.01, 2*1-sided exact *p* = 0.01, adjusted z = −2.67, R^2^ = 0.98), which was lower in 1PP (72.34 ± 11.02) than in 3PP (110.84 ± 11.26). Additionally, we compared the t_1_ of the two sessions to control for any differences between the baselines, considering they were recorded 10 weeks apart. We can confirm that no significant differences emerged, even considering the two groups separately.

**Figure 4 ijerph-19-06340-f004:**
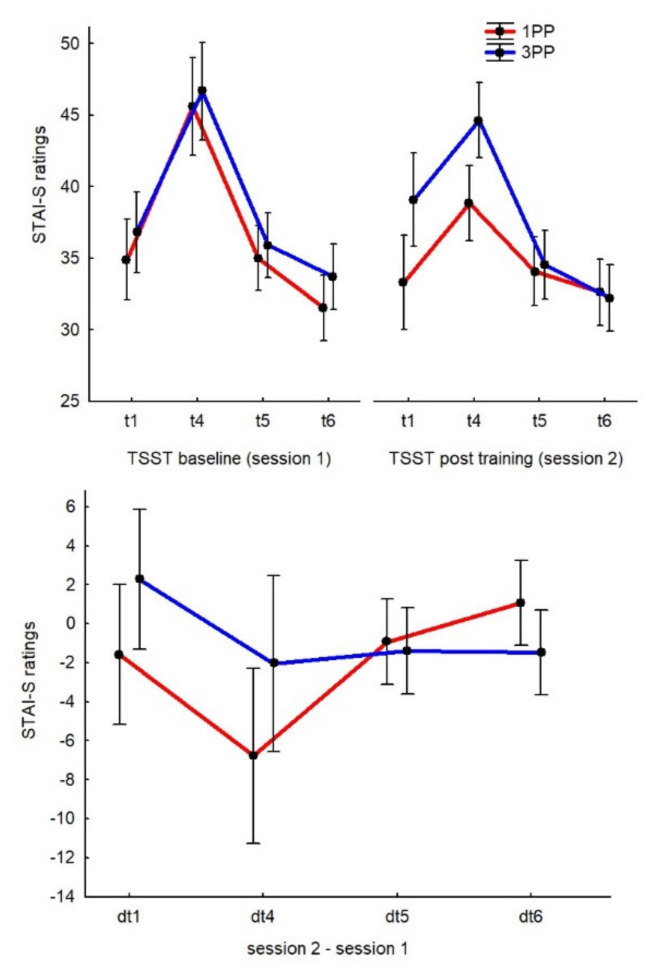
Graphs of the STAI-S results. The top chart represents the two sessions separately and the bottom chart represents the differences between the sessions. The time points (*t*) and delta time points (dt) of the STAI-S measurements are represented as in Figure 3.

**Figure 5 ijerph-19-06340-f005:**
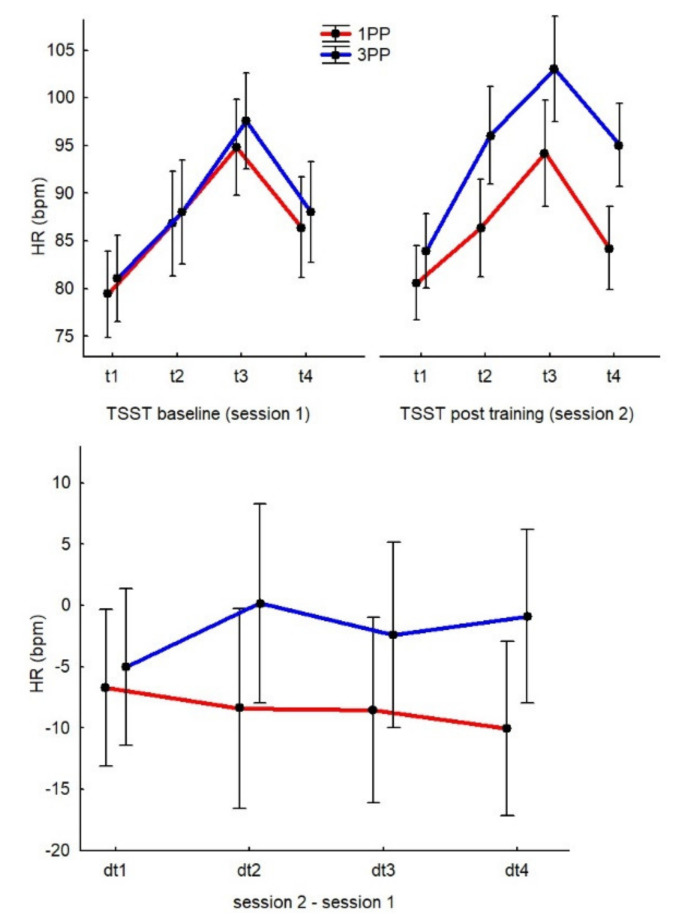
Graphs of the HR results. The top chart represents the two sessions separately and the bottom chart represents the differences between sessions. In the top graph, the time points (t, averaged) of the HR measurements are represented as follows (also see Figure 1 as a reference): the 5-min recording from the pre-stress phase (t1), the 10-min recording from the speech preparation phase (t2), the 5-min recording during the speech delivery phase (t3) and the 5-min recording during the math task (t4) for each session. In the bottom graph, the delta time points (dt) represent the data from Session 2 minus the data from Session 1 (TSST post-training − TSST baseline, e.g., dt1 = t1 of Session 2—t1 of Session 1).

**Table 1 ijerph-19-06340-t001:** Statements and results of the online and offline questionnaires. The columns, from left to right, indicate the phase during which the questionnaire was administered (online or offline; static or dynamic phase), the statement number (for example, s1), the underlying concepts (not disclosed to participants) and the actual statements. The z-score values (ipsatized; average ± SE) that were used in the analysis are shown for the first person perspective (1PP) and third person perspective (3PP) groups, while in parenthesis the non-ipsatized results (on the 1–7 Likert scale) are expressed as the average ± SE. For the dynamic phase, the displayed results are the average of the three repetitions. In the last two columns, * highlights a significant difference between the results and when *p* was significant, the effect sizes are shown.

				1PP	3PP	*p* Value	R^2^
**Online Questionnaire**							
Static Phase	s1	Body ownership	I feel as if I am looking at my own body	3.91 ± 0.26 (5.92 ± 0.22)	0.17 ± 0.13 (1.88 ± 0.16)	<0.01 *	0.67
	s2	Body ownership control	I feel as if the virtual body belongs to another person	0.45 ± 0.24 (2.46 ± 0.26)	4.40 ± 0.27 (6.11 ± 0.23)	<0.01 *	0.66
Dynamic Phase	s1	Body ownership	I feel as if I am looking at my own body	3.53 ± 0.28 (5.53 ± 0.27)	0.48 ± 0.17 (2.19 ± 0.23)	<0.01 *	0.57
	s2	Body ownership control	I feel as if the virtual body belongs to another person	1.13 ± 0.26 (3.14 ± 0.28)	3.37 ± 0.33 (5.08 ± 0.31)	<0.01 *	0.34
	s3	Agency	The virtual body moves just as I want, as if I am controlling it	2.63 ± 0.24 (4.64 ± 0.22)	0.72 ± 0.20 (2.43 ± 0.25)	<0.01 *	0.42
	s4	Agency control	I feel as if the virtual body is controlling my will	0.36 ± 0.22 (2.37 ± 0.25)	0.49 ± 0.21 (2.20 ± 0.26)	0.45	
**Offline Questionnaire**							
	s5	Located	During the experiment, I felt as if my body was located where I saw the virtual body to be	1.88 ± 0.23 (4.69 ± 0.26)	1.26 ± 0.27 (3.19 ± 0.28)	0.04 *	0.07
	s6	Ownership	During the experiment, I felt that the virtual body was my own body	1.73 ± 0.28 (4.53 ± 0.31)	0.69 ± 0.23 (2.61 ± 0.26)	0.01*	0.13
	s7	Standing	During the experiment, I felt that I was standing upright	1.81 ± 0.25 (4.61 ± 0.26)	0.62 ± 0.27 (2.53 ± 0.28)	<0.01 *	0.18
	s8	My movements	During the experiment, I felt that the leg movements of the virtual body were my movements	1.58 ± 0.23 (4.38 ± 0.24)	0.46 ± 0.23 (2.38 ± 0.27)	<0.01 *	0.20
	s9	Agency	During the experiment, I felt that the leg movements of the virtual body were caused by my movements	1.35 ± 0.24 (4.15 ± 0.26)	0.23 ± 0.24 (2.15 ± 0.28)	<0.01 *	0.17
	s10	Ownership control	During the experiment, I felt that the virtual body belonged to someone else	0.27 ± 0.31 (3.07 ± 0.24)	3.23 ± 0.42 (5.15 ± 0.34)	<0.01 *	0.35
	s11	Effort	I felt that I had to give extra physical effort when the virtual body was running	0.81 ± 0.29 (3.61 ± 0.33)	0.15 ± 0.12 (2.07 ± 0.18)	0.07	
	s12	Vection	I felt that I was moving through space rather than the world moving past me	2.31 ± 0.26 (5.11 ± 0.28)	1.85 ± 0.36 (3.76 ± 0.36)	0.29	
	s13	Walking	I felt that I was walking	1.85 ± 0.23 (4.65 ± 0.27)	1.62 ± 0.31 (3.53 ± 0.33)	0.40	
	s14	Dragged	I felt that I was being dragged	−0.30 ± 0.16 (2.50 ± 0.22)	0.46 ± 0.21 (2.38 ± 0.23)	<0.01 *	0.12
	s15	Sliding	I felt that I was sliding	−0.41 ± 0.18 (2.38 ± 0.24)	1.39 ± 0.40 (3.30 ± 0.41)	<0.01 *	0.24

**Table 2 ijerph-19-06340-t002:** Descriptive values for the sAA measurements for each time point, according to group. For both groups and all time points in both sessions, the descriptive statistics of the sAA results from during the TSST are displayed as the average (SE). All delta time points (session 2−session 1) are also shown.

	Session 1		Session 2		Session 2−Session 1	
1PP	t_1_	51.88 (9.53)	t_1_	40.00 (8.39)	dt_1_	−11.88 (5.56)
			t_0_	114.11 (17.44)		
	t_4_	134.03 (13.28)	t_4_	72.34 (11.02)	dt_4_	−61.69 (9.13)
	t_5_	108.57 (12.09)	t_5_	56.19 (10.32)	dt_5_	−52.38 (7.62)
	t_6_	75.57 (10.11)	t_6_	44.03 (8.12)	dt_6_	−31.53 (6.74)
3PP	t_1_	53.73 (8.94)	t_1_	64.11 (9.73)	dt_1_	10.38 (6.04)
			t_0_	88.26 (16.40)		
	t_4_	122.07 (19.33)	t_4_	110.84 (11.59)	dt_4_	−11.23 (5.14)
	t_5_	119.92 (11.32)	t_5_	100.88 (11.21)	dt_5_	−19.03 (10.06)
	t_6_	110.84 (11.26)	t_6_	80.42 (9.90)	dt_6_	−30.42 (11.76)
All	t_1_	52.80 (6.47)	t_1_	52.05 (6.58)	dt_1_	−0.75 (4.35)
			t_0_	101.19 (11.99)		
	t_4_	128.05 (11.64)	t_4_	91.59 (10.62)	dt_4_	−36.46 (6.33)
	t_5_	114.25 (8.24)	t_5_	78.53 (8.16)	dt_5_	−35.71 (6.67)
	t_6_	93.21 (7.89)	t_6_	62.23 (6.83)	dt_6_	−30.98 (6.71)

**Table 3 ijerph-19-06340-t003:** Descriptive values for the STAI-S measurements for each time point, according to group. For both groups and all time points in both sessions, the results of the STAI-S measurements from during the TSST are displayed as the average (SE). All delta time points (session 2 − session 1) are also shown. The last column shows *p* values when dt4 (immediately after the math task of the TSST, which is supposed to be the peak of the stress response) was compared to the other delta time points. * highlights a significant difference between the time point indicated in the row and dt_4_.

	Session 1		Session 2		Session 2 − Session 1		dt_4_
1PP	t_1_	34.88 (1.52)	t_1_	33.30 (1.50)	dt_1_	−1.57 (1.45)	0.02 *
			t_0_	43.73 (1.72)			
	t_4_	45.61 (1.70)	t_4_	38.84 (1.19)	dt_4_	−6.76 (2.04)	---
	t_5_	35.00 (1.00)	t_5_	34.07 (1.19)	dt_5_	−0.92 (1.23)	0.01 *
	t_6_	31.53 (1.13)	t_6_	32.61 (1.23)	dt_6_	1.07 (1.27)	<0.01 *
3PP	t_1_	36.80 (1.28)	t_1_	39.07 (1.74)	dt_1_	2.26 (2.07)	<0.01 *
			t_0_	44.76 (1.52)			
	t_4_	46.69 (1.69)	t_4_	44.65 (1.41)	dt_4_	−2.03 (2.41)	---
	t_5_	35.92 (1.23)	t_5_	34.53 (1.18)	dt_5_	−1.38 (0.95)	0.04 *
	t_6_	33.69 (1.15)	t_6_	32.23 (1.07)	dt_6_	−1.46 (0.83)	0.03 *
All	t_1_	35.84 (0.99)	t_1_	36.19 (1.21)	dt_1_	0.34 (1.28)	
			t_0_	41.75 (1.00)			
	t_4_	46.15 (1.19)	t_4_	41.75 (1.00)	dt_4_	−4.40 (1.60)	
	t_5_	35.46 (0.78)	t_5_	34.30 (0.83)	dt_5_	−1.15 (0.77)	
	t_6_	32.61 (0.81)	t_6_	32.42 (0.80)	dt_6_	−0.19 (0.77)	

**Table 4 ijerph-19-06340-t004:** Descriptive values for the HR measurements for each time point (averaged), according to group. For both groups and all time points in both sessions, the results are displayed as the average (SE) in bpm. All delta time points (session 2 − session 1) are also shown.

	Session 1		Session 2		Session 2 − Session 1	
1PP	t_1_	79.42 (2.36)	t_1_	72.72 (2.54)	dt_1_	−6.69 (3.62)
	t_2_	86.82 (2.66)	t_2_	78.43 (2.80)	dt_2_	−8.39 (4.14)
	t_3_	94.83 (2.34)	t_3_	86.31 (3.34)	dt_3_	−8.52 (4.03)
	t_4_	86.43 (2.33)	t_4_	76.38 (2.34)	dt_4_	−10.04 (3.39)
3PP	t_1_	81.10 (2.13)	t_1_	76.07 (1.53)	dt_1_	−5.03 (2.66)
	t_2_	88.01 (2.77)	t_2_	88.17 (1.97)	dt_2_	0.16 (3.95)
	t_3_	97.54 (2.65)	t_3_	95.15 (2.03)	dt_3_	−2.38 (3.47)
	t_4_	88.05 (2.88)	t_4_	87.16 (1.97)	dt_4_	−0.89 (3.65)

## Data Availability

The data presented in this study are available upon request from the corresponding author. The data are not publicly available, as requested and stated by the ethical committee of the Tohoku University Graduate School of Medicine.

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
