# Peer review of "Neuroendocrine Response and State Anxiety Due to Psychosocial Stress Decrease after a Training with Subject’s Own (but Not Another) Virtual Body: An RCT Study"

_ijerph, 2022, doi:10.3390/ijerph19106340_

Round 1

Reviewer 1 Report

Thank you for the opportunity to review this paper. The objective of this study was to investigate the psychosocial stress response after a virtual training performed exclusively by a virtual body, while the person was not performing any actual movement and the conclusion of the study is that watching a training performed by own virtual body induce neuroendocrine changes. The effect was less important on the anxiety scales.

The theoretical background is well documented, the methods section describes the study in detail so the study can be replicate and the conclusions are supported by the data.

Reviewer 2 Report

The current study adopted a RCT study to test neuroendocrine and psychological responses when seeing a moving virtual body in first and third person perspectives. It has great application value in some fields, I believe. However, there are still some concerns should be addressed in this manuscript.

  1. The results displayed in Abstract is so brief. More details (e.g., heart rate) should be provided.
  2. The significance test of demographic information for Table S2 between two groups should be performed. Variables with significant differences should be controlled as covariables in data analyses.
  3. The limitations should be placed in Discussion, not Conclusion.

Reviewer 3 Report

Thank you for the opportunity to review this interesting paper. The results are both, inspiring and scary. The study results support huge power of virtual illusion which might be used as therapeutic mean, but also raises a lot of philosophical questions that exceed the scope of this paper.

The manuscript is well written, and steps of the study protocol and experiments comprehensively described. Authors provide detailed statistical analysis and concentrated discussion. Due to high amount of text information, it is a little but hard to follow all the presented results, thus, I would suggest including some tables in the main text. It is uncomfortable when all the information is provided in supplements. I would also suggest explaining abbreviations in the figures because it should be possible to understand figures without reading a text.

The main limitation of the study is a sample. The sample consisted of young and healthy adults that are actually capable for adequate physical activity. Please provide arguments, why such a sample was chosen. In the introduction section authors stay that virtual physical activity might be beneficial for those with physical frailty, thus it is not clear, why healthy people were chosen and strict exclusion criteria were set.  

Another limitation is repetition of TSST. Although authors provide explanations and demonstrate efforts to make two sessions comparable, one very important element – novelty of experimental situation - is lost in second session, that may result in lower stress response. As I understand, authors did not try to balance effect of order by dividing experimental groups in subgroups (1/2 -TSST alone when VR+TSST; ½ VR+TSST then TSST alone). I would recommend discussing effect of the order in discussion section.

As a health specialist, who usually advocates for physical activity, I also would like to see more discussion about pros and cons of virtual reality training. The finding, that VR may induce changes in mood or even endocrine response, do not necessarily mean, that it is beneficial. Alcohol or psychotropic drugs do exactly the same – change mood and induce biological response. Discussion of possible side effects of VR is recommended.

Author Response

This manuscript is a resubmission of an earlier submission. The following is a list of the peer review reports and author responses from that submission.

Round 1

Reviewer 1 Report

This manuscript describes results from a RCT conducted with a sample of young and healthy Japanese adults, who participated in a virtual reality physical exercise training. Participants were allocated to one of two groups, based on the perspective from which the training was conducted. For a duration of 30 minutes, group one was presented with a running virtual avatar from a first-person perspective, while group two was presented with an avatar from a third-person perspective. Central comparisons made were for physiological (salivary alpha amylase, primary outcome measure) and psychological responses (stait anxiety, perceived stress). Psychosocial stress was experimentally induced via a well-established and widely used stress test, that is the Trier Social Stress Test, which is known to reliably induce a strong psychobiological stress response in humans. It was hypothesized that performing the virtual training from a first-person perspective would lead to a stronger decrease in physiological and psychological stress measures, than performing this training from a third-person perspective. To investigate whether the effect of the training on stress levels (physiological and psychological) differed between groups, the Trier Social Stress Test was conducted with both groups at two points in time, that is, during a visit 10 weeks before the virtual training, which served as a baseline, and during a second visit, right after the virtual training. A significant difference between groups was found for the primary outcome measure concentration of alpha-amylase right after the unexpected math task (difference between baseline and endline measure at timepoint 4) of the TSST, with alpha amylase levels being significantly lower in the group which participated in the virtual training from a first-person perspective as compared to the group which participated in the training from a third-person perspective.

This is a timely and interesting study in which the possible strengths of VR are investigated; that is if VR can transfer physical benefits in subjects to the real world. This topic has importance to the field of VR research. Strengths of this study include the RCT design, the well-chosen exclusion criteria. The paper is well-written. My main concern includes the repeated usage of the TSST.

Major revisions

  1. Repeated usage of the TSST: My main concern about the methodology of the paper is the repeated usage of the TSST. This is, in general, not possible to induce reliable effects because of a learning experience of subjects. The authors write on page 5 that they took measures to avoid habituation and learning effects when using the TSST twice. One of those measures was to modify the details of both speech and mathematical task when the TSST was performed for a second time. Please provide more detail on the reasoning behind exchanging the mock job interview for asking participants to publicize their home town to a panel of marketing experts. Why should this have the same effect on the neuroendocrine stress response as the mock job interview? Is this method validated by previous research? The authors need to provide references of research that justifies the repeated usage of TSST.

  2. Primary outcome measure (salivary alpha amylase): In this study, salivary alpha amylase was the primary outcome measure as it is considered a biomarker for a psychosocial stress response. Nevertheless, cortisol measurement has been used most widely in stress research. With the information provided, it remained unclear, why the choice was made to diverge from this commonly used measure. Please provide more detailed information on the reasoning behind this choice, to support the choice of your primary outcome measure.

  3. Randomization: Please provide more detail on the randomization device used for this study. The authors mention “Group assignment was managed by a co-experimenter using a simple randomization 1 (experimental):1 (control) ratio, with the allocation of participants to each arm based on the order of entry into the study.” This short description raised the question of why you chose to do it this way, given that there are less error prone methods for randomization available, as for instance the usage of a computerized random numbers generator.

  4. Relatedly, the author mention a co-experimenter assigned participants to groups. Could you provide information on the measures you took to blind treatment allocation as much as possible? This could help rule out experimenter effects, which threaten the validity of the study.

  5. Pre-registration: Based on the current information it cannot be concluded with certainty that the study was pre-registered. Could the authors provide information on whether the data was pre-registered and if so, where the pre-registration plan could be found? This is important to verify if no important changes were made after commencement of recruitment of participants.

Minor revisions

  1. Baseline similarity of groups: The link provided to access supplementary material on page 18 returns an error message (page not found), so it was not possible to check that participants were homogenously distributed across groups by means of randomization, or to check any other supplementary materials.

  2. Attrition: This experiment comprised two appointments for participants. Information on attrition from appointment one to appointment two was not provided. I advise to include a flow chart of the participants. I assume that the flow chart of participants is given in the supplementary material as the authors state they follow CONSORT guidelines. However, I was unable to access this material.    

Reviewer 2 Report

Overall, an interesting study. I do, however, have several reservations. Unfortunately, those are mainly methodological.
Firstly, however, I don’t feel like the title illustrates the content of the study itself. The study makes an argument that observing a running model in immersive virtual reality from a 1PP but not 3PP leads to reductions in alpha-amylase and state anxiety.

Furthermore, as the model was not personalised to each participant, I think it is incorrect to use the word ‘avatar’; an avatar is supposed to be a representation of a particular individual. Instead ‘virtual body’ should be used as used by the authors. I would recommend changing the title. Especially as the participants did not participate in training, they merely observed it.

The main methodological weakness is the procedure itself. There is no clear rationale for why the baseline measurements were taken so far in advance – 10 weeks.
Please explain why the markers were not collected simply before and after the VR task? It does not seem prudent to not record all of these markers immediately prior to the task? What is the usefulness of this information from 10 weeks ago? The procedure should have been: baseline, then VR task, then post VRtask measurements, so that clear comparison can be drawn. In light of not knowing what happened in those 10 weeks and what the actual stress levels, sAA etc. were prior to the task, it is not a fair conclusion that it was the task itself that created these differences.  The rationale for this procedure needs to be provided or perhaps the authors may reconsider how they present their findings.

Furthermore, the participants were expected to observe the body for a very long period of time. While 30 minutes does not seem long, it is very long to observe something that is repetitive and potentially not interesting, while simultaneously deprived of other input such as sound. As such, is it possible that the participants closed their eyes? Especially those in the 1PP may be more likely to do so, as they are in a more resting position than those who had to sit up straight in 3PP? If this was to be the case, those participants could potentially be resting, which would explain the differences observed. Newer VR headsets allow for eyetracking.

Additionally, performing the TSST the second time means that the participants already performed it once, so may not react to the task as much as they did before. Additionally, the topics of the two speeches are so different that this review does not consider them comparable (ideal job vs. publicise hometown).

Next, the green ground displayed in the VR environment is uniform – as such there would be no indicators of movement forward. In my experience people who remain stationary for some time in VR while embodying a model that moves experience that movement; usually in negative manner (dizziness, motion sickness). While there was a measure of ownership, there is only one statement on vection.

Lastly, there is no true control group. Both groups were experimental, a true control group would be simply sat down for the same amount of time and had the same measurements taken without any intervention. As both groups were asked to observe the exercise either from 1PP or 2PP, they both are experimental groups. I would recommend referring to the study as a pilot, rather than an RCT, and to overall use more careful language

Some additional thoughts:

Throughout the manuscript the authors refer to the participants having agency over the virtual body, as that is not the case, they are merely observers, this should be changed.

In the discussion, the lack of true control group should be acknowledged, discussed, and suggestions for improvement suggested.

P1, line 19: correct to “this evidence”

P2, lines 58-61: The relationship between “obesity” and risk of cardiovascular diseases (amongst others) is disputed, I would recommend removing it.

P2 the introduction needs to create a much stronger case for the impact of VR (lines 66-71) this section needs to be much more detailed to be fully convincing.

P 2, line72: correct ‘evidences’ to ‘evidence’

The exercise is referred to as ‘moderate’ in line 257, but then as ‘acute’ in line 537.

Line 543 correct “his/herself” to “themselves”

Round 2

Reviewer 1 Report

The manuscript has improved after the revisions. However, the authors did not convince me in the rationale of using the TSST twice. In addition, randomization has not been carried out properly and I just noticed when I was rereading the manuscript that baseline was taken 10 weeks prior to the experiment. Specifically for acute stress this can introduce major flaws in the findings.  

Reviewer 2 Report

1. I appreciate the corrected title.

2. I am absolutely certain that I read the correct manuscript. Not only do the authors use the word “avatar” repeatedly in first version of the manuscript but also in the second version, and even in the response letter. Let me re-iterate: you are using the word “avatar” incorrectly. Simply embodying a model from 1PP does not make the model an avatar.

3. While I appreciate that the authors were trying to address the limitations of the TSST, I do not find it a sufficient justification of this to allow for design where the conclusions have to be drawn with a great deal of carefulness. Perhaps authors should investigate other methods of measuring or inducing stress.

4. I have no intention to provide a link, a simple google search will provide you with VR headsets that allow eye tracking. Speaking may mean that authors are preventing the participants from dozing off, but this does not mean that they are paying attention to what is displayed to them in VR or that they have their eyes open. If the questions were displayed in VR would ensure that they saw them (given that their eyes were open). As such, I do not believe that the authors can claim to have measured visual attention. Participants can still answer questions even if they keep their eyes closed.

5. Fair

6. The horizon would not have been visible to the 1PP participants because they are looking down. The image provided shows that the ground is indeed meadow-green but it does not seem to have any distinctive features that indicate movement, it is all solid green colour.

7. I am not convinced by this response. You have omitted a very important component of an RCT and to call this study an RCT is misleading.

11. You have missed the point of my comment, it is not about semantics. ‘Overweight’ is just as bad as saying ‘obese’. Over what weight exactly? This is the weight-stigma language that really needs to stop in science.